# Study on Changes in Gut Microbiota and Microbiability in Rabbits at Different Developmental Stages

**DOI:** 10.3390/ani14121741

**Published:** 2024-06-08

**Authors:** Chong Fu, Yue Ma, Siqi Xia, Jiahao Shao, Tao Tang, Wenqiang Sun, Xianbo Jia, Jie Wang, Songjia Lai

**Affiliations:** 1College of Animal Science and Technology, Sichuan Agricultural University, Chengdu 611130, China; 2Farm Animal Genetic Resources Exploration and Innovation Key Laboratory of Sichuan Province, Sichuan Agricultural University, Chengdu 611130, China

**Keywords:** rabbits, feces, 16s rRNA sequencing, microbial species, microbiability

## Abstract

**Simple Summary:**

In recent years, research has shown that animals’ gut microbiota can regulate various physiological functions such as nutrition, metabolism, and immunity, and play an indispensable role in maintaining host gut health. At present, there are few reports on the establishment of rabbits’ gut microbiota and its relationship with the host. To clarify the colonization of rabbits’ gut microbiota, the changes in species during development, and the relationship between gut microbiota and rabbits’ growth and development, this study determined the 16s rRNA of gut microbiota in rabbits of different age groups and preliminarily explored the changes in the types of gut microbiota during the colonization process. Six amplicon sequence variants (ASVs) that significantly affect body weight were obtained. The research results provide a firm theoretical basis for maintaining gut health and constructing new methods for rabbit breeding.

**Abstract:**

This study used feces from 0-day-old (36 rabbits), 10-day-old (119 rabbits), and 60-day-old (119 rabbits) offspring rabbits and their corresponding female rabbits (36 rabbits) as experimental materials. Using 16s rRNA sequencing, the study analyzed the types and changes of gut microbiota in rabbits at different growth and development stages, as well as the correlation between gut microbiota composition and the weight of 60-day-old rabbits. All experimental rabbits were placed in the same rabbit shed. Juvenile rabbits were fed solid feed at 18 days of age and weaned at 35 days of age. In addition to identifying the dominant bacterial phyla of gut microbiota in rabbits at different age stages, it was found that the abundance of *Clostridium tertium* and *Clostridium paraputrificum* in all suckling rabbits (10-day-old) was significantly higher than that in rabbits fed with whole feed (60-day-old) (*p *< 0.05), while the abundance of *Gram-negative bacterium cTPY13* was significantly lower (*p *< 0.05). In addition, Fast Expected Maximum Microbial Source Tracing (FEAST) analysis showed that the contribution of female rabbits’ gut microbiota to the colonization of offspring rabbits’ gut microbiota was significantly higher than that of unrelated rabbits’ gut microbiota (*p *< 0.05). The contribution of female rabbits’ gut microbiota to the colonization of gut microbiota in 0-day-old rabbits was significantly higher than that to the colonization of gut microbiota in the 10- and 60-day-old rabbits (*p *< 0.05). Finally, the correlation between gut microbiota composition and body weight of 60-day-old rabbits was analyzed based on a mixed linear model, and six ASVs significantly affecting body weight were screened. The above results provide important theoretical and practical guidance for maintaining gut health, improving growth and development performance, and feeding formulation in rabbits.

## 1. Introduction

The growth and development of livestock is influenced by various factors, including genetic factors, nutritional levels, feeding management, and gut microbiota [1,2]. The gut microbiota is a designated microbial community that helps digest food and absorb nutrients, promotes immune system development, regulates metabolism, and prevents the growth of harmful bacteria [3,4,5]. In many animals, microorganisms are an important component of the host ecosystem and metabolic capacity, becoming a fundamental element for the host’s survival and sustained existence in time and space [6]. In recent years, the role of gut microbiota in the production performance of livestock has received increasing attention. In rabbits, gut microbiota can predict their growth performance and affect muscle metabolism, thereby affecting rabbit meat quality [7,8]. Yang et al. [9] found that the gut microbiota explained nearly 11% of weight changes, indicating that gut microbiota had a significant impact on the mature weight of rabbits. Fang et al. [10] detected 50 operational taxonomic units (OTUs) significantly associated with the weaning weight of meat rabbits and found that members of the *Ruminococcaceae* family were involved in the degradation of indigestible fiber and polysaccharides in the diet, as well as the production of butyrate, which was important for improving weaning weight.

Studies have shown that in mammals the gut microbiota is usually shaped by the mother at birth and can be transmitted from the mother’s placenta, uterus, and vagina to the baby [11]. The structure of the gut microbiota has a significant impact on the development of the offspring’s gut system and immune system construction [12]. A recent study has shown that based on a comparison of hen feces and gut microenvironment the microbiota is inherited from hens to chicks, and it was preliminarily inferred that in poultry the gut microbiota of the female can also be transmitted to the next generation [13]. Savietto et al. found [14] that rabbit younglings were born with a sterile digestive tract but that then it got progressively colonized by the microbiota of the nursing mother as a result of the younglings entering into contact with or ingesting the maternal droppings present in the nest. In addition, the gut microbiota is further influenced by environmental factors such as host nutrition, lifestyle, and immune status during the growth and development of the host, leading to changes in them [15,16]. María et al. [17] found that host genetics shape the overall microbial diversity in the cecum of rabbits, and a large portion of taxa were influenced by host genetics or environmental factors.

This study utilized 16s rRNA sequencing technology, FEAST, and mixed linear models to reveal the diversity and differences in microbial composition at different growth and development stages of rabbits. The study investigated the impact of female rabbits’ gut microbiota on the establishment of offspring gut microbiota, compared the similarity of female and offspring rabbits’ gut microbiota, quantitatively analyzed the contribution ratio of female rabbits’ to offspring rabbits’ gut microbiota, and explored the correlation between rabbits’ microbial community strength, relative microbial abundance, and body weight. Microbial biomarkers significantly associated with body weight were screened. The research results can provide a firm theoretical basis for maintaining gut health and devising new methods for rabbit breeding.

## 2. Materials and Methods

### 2.1. Ethics Statement

All experiments in the current work involving animals were performed under the direction of the Institutional Animal Care and Use Committee from the College of Animal Science and Technology, Sichuan Agricultural University, China (DKY-B2019302083).

### 2.2. Animal and Experimental Design

This experiment was conducted at the Rabbit Farm of Sichuan Agricultural University. In total, 36 litters of female Tianfu black rabbits with similar delivery dates (±1 day) were selected as the experimental animals. Feed supply (details are shown in Appendix A) and feeding management were consistent, and the experimental rabbit houses were naturally lit and ventilated by negative pressure fans. During the experiment, feed was fed daily in the morning and evening, and water was provided by an automatic drinking water device. Additionally, there was a device for mechanical defecation removal.

After the female rabbits gave birth, there was a total of 296 offspring in 36 litters of female rabbits. The juvenile rabbits were fed along with the female rabbits and weaned at 35 days of age. After weaning, 3 rabbits were fed in each cage, and weaned rabbits were fed until 60 days of age. The 0-day-old rabbits were not breastfed; 10-day-old rabbits were only fed with breast milk, and 60-day-old rabbits were fed pellet feed. Three different age groups of gut contents and feces were sampled to explore the effects of different age groups and feeding conditions on the types of gut microorganisms. At 0 days of age, a rabbit was randomly selected and euthanized, and rectal contents were collected as samples for 1 day of age. Feces were collected from experimental rabbits on the day of delivery and at 10- and 60-days-old. On the day for collecting feces, each experimental rabbit was raised separately in a cage, with clean kraft papers laid on the bottom of the cage. When collecting feces, the fecal skin was peeled off, and the middle feces of each uncontaminated particle were placed in a frozen storage tube. The collected fecal samples were stored in liquid nitrogen. The number of samples was as follows: 36 female rabbits, 36 rabbits at 0 days old, 119 rabbits at 10 days old, and 119 rabbits at 60 days old. Of the 296 offspring of the 36 female rabbits, due to reasons such as being euthanized at 0 days old, lack of milk, or illness, only 119 rabbits survived at 60 days old, so 119 corresponding rabbits were selected at 10 days old. All female rabbit information is shown in Appendix A, and all offspring rabbit information is shown in Appendix A.

### 2.3. DNA Extraction and Sequencing

Fecal genomic DNA was extracted by the magnetic bead method. The final DNA concentration and purity were determined using a NanoDrop 2000 UV–vis spectrophotometer (Thermo Scientific, Wilmington, NC, USA), and DNA quality was checked by 1% agarose gel electrophoresis. Subsequently, the samples were diluted to 1 ng/μL with sterile water. The corresponding region of the primers was 16s V3–V4 region, with primer sequences CCTAYGGGRBGCASCAG and GGACTACNNGGGTATCTAAT. All PCR mixtures were made with with 15 µL Phusion^➅^ High Fidelity PCR Master Mix, 0.2 µM primers and 10 ng genomic DNA template. The first denaturation was performed at 98 °C for 1 min, followed by 30 cycles at 98 °C (10 s), 50 °C (30 s), and 72 °C (30 s), and finally maintained at 72 °C for 5 min. PCR products were detected by electrophoresis using agarose gel with 2% concentration. The qualified PCR products were purified by magnetic beads, quantified by enzyme label, and mixed in equal amounts according to the concentration of the PCR products. After full mixing, the PCR products were detected by 2% agarose gel electrophoresis. For the target strip, the universal DNA purification and recovery kit was used. A DNA library-building kit was used for library construction, and the constructed library was quantified by Qubit and Q-PCR. After passing the qualification test, it was sequenced using NovaSeq PE 250.

### 2.4. Bioinformatics Analysis

FLASH was used to concatenate the reads of each sample, and FASTP (0.23.1) software was used to filter and process them. The DADA2 module in QIIME2 (2022.2) software was used for noise reduction to obtain the final ASVs. They did not cluster sequences based on a distance-based threshold, thus allowing for a finer taxonomic classification level compared to operational taxonomic units (OTUs). The feature table module of QIIME 2 was used to obtain the feature table. The feature classifier module of QIIME2 software was used for species annotation, and the database used was Silva138.1. Then, QIIME 2 software was used to calculate the Shannon index, and significant differences between groups were tested using Tukey. Linear discriminant analysis effect size (LEfSe) was based on the Galaxy Server 2.0 of Hutten Power Laboratory, which uploads the feature table obtained from Qiime2 directly to the web for analysis. The prediction calculation of Tax4Fun was completed in the R programming language, and all parameters were default parameters. FEAST analysis was based on a feature table, with 3000 reads per sample and 1000 expected maximum iterations, all of which were the default settings for FEAST.

### 2.5. Statistical Analysis

For weight characteristics, the following four models were fitted:(1)y=Xb+e
(2)$$y=Xb+Z_{2}m+e$$
(3)$$y=Xb+Z_{1}a+e$$
(4)$$y=Xb+Z_{1}a+Z_{2}m+e$$
Among the models, *y* is body weight, *b* is a fixed effect vector, *a* is a random breeding value vector, *m* is a random microbial vector, *e* is a random residual, and *X*, *Z*_1_, and *Z*_2_ are the correlation matrices corresponding to *b*, *a*, and *m*. The random effects are assumed to follow the distribution: $a∼N\left(0,A\sigma_{m}^{2}\right),\;m∼N\left(0,M\sigma_{m}^{2}\right)$ and $e∼N\left(0,I\sigma_{e}^{2}\right)$, where $\sigma_{e}^{2},\;\sigma_{m}^{2},$ and $\sigma_{m}^{2}$ correspond to genetic variance, microbial variance, and residual variance, respectively. *I* is the identity matrix, *A* is the pedigree matrix of the rabbits, and *M* is the microbial association matrix, defined as $M=\frac{Z_{3}Z_{3}^{\prime }}{k}$, where the dimension of *Z*_3_ is n×k, *n* is the number of rabbits with relative microbial abundance, and *k* is the number of ASVs. Pedigree information for the 119 rabbits can be found in Appendix A. Each element ${P_{i}}_{j}$ in the *Z*_3_ matrix is the normalized abundance of the *j*th ASV corresponding to the *i*th individual. The normalization formula is as follows:(5)$$Z_{3_{i_{j}}}=\frac{log\left(P_{i_{j}}\right)-\sqrt{log{\left(P_{i_{j}}\right)}_{j}}}{sd{\left(log\left(P_{i_{j}}\right)\right)}_{j}}$$
Fixed effects include gender, cage size, and birth weight, based on Bayesian GIBBFSF90 [18]. Four models were fitted to estimate the degree of fit of microbial effects, and goodness of fit was compared between models based on the deviation information criterion (DIC).

ASVs with a relative abundance of 1% were screened, and 1136 were obtained. Regression analysis was performed on each of these ASVs, and the associated *p*-values were calculated. The BLUP method of AIREMLF90 software (64bit AMD) was used to fit the model. In addition, the iteration and parameter file modification were based on Python. The regression coefficients of the fitted ASV covariates were estimated, and their standard errors converted into Z-scores. The chi-square test was used to calculate the *p*-values. The process of statistical analysis was completed using the SciPy library.

## 3. Results

### 3.1. Data Evaluation

16s rRNA gene sequencing was performed on the fecal microbiota of female rabbits and their 0-, 10-, and 60-day-old rabbits. A total of 27,351,012 original sequences were obtained from 310 samples. After a series of quality control measures and bioinformatics processing, 26,331,830 valid sequences were obtained. A new dataset was formed with an average of 84,941 sequences and a Q20 of 98.53% for subsequent analysis (Figure 1A, Appendix A). According to the Venn Diagram, the 0-, 10-, and 60-day-old rabbit and female rabbit groups contain 269 core ASVs, with 10,986 ASVs, 6210 ASVs, 6090 ASVs, and 3801 ASVs unique to the 0-, 10-, and 60-day-old rabbit and female rabbit groups, respectively (Figure 1B). To determine whether the sampling depth was sufficient to explain the rabbits’ fecal microbiota, rarefaction and Shannon curves were generated for the ASV quantity of each group. The results showed that as the feature sequence increased, the final curve tended to flatten, indicating that there were sufficient sequences to represent each microbiota, ensuring the adequacy and accuracy of the analysis (Figure 1C,D). In addition, we found that there were differences in the curve levels between different groups, especially the 10-day-old rabbit group, indicating potential differences in the total number and diversity of microbial populations between different groups.

### 3.2. Composition and Comparison of Gut Microbiota in Female Rabbits and Their Different Aged Juvenile Rabbits

To reveal the differences in gut microbiota between female rabbits and their 0-, 10-, and 60-day-old rabbits, we conducted alpha diversity and beta diversity analyses on sequencing data. In the alpha diversity analysis, the Shannon index results showed significant differences in gut microbiota among different-day-old rabbits, with significant differences between groups of at least *p *< 0.01; the richness of gut microbiota first decreased and then increased (Figure 2A). In the beta diversity analysis, PCoA results showed significant differences in the composition of gut microbiota among rabbits of different ages; the gut microbiota composition of 60-day-old rabbits overlapped with that of the female rabbits, indicating that the gut microbiota composition of 60-day-old rabbits was very similar to that of the female rabbits (Figure 2B).

In addition, we conducted species abundance analysis on the gut microbiota of female rabbits and their 0-, 10-, and 60-day-old rabbits, identifying a total of 40 phyla, 86 classes, 196 orders, 323 families, 673 genera, and 420 species. At the level of the phylum, there were significant changes in the composition of gut microbiota in female rabbits and their 0-, 10-, and 60-day-old rabbits. The dominant phyla of gut microbiota in female rabbits included *Firmicutes* (52.81%), *Bacteroidetes* (24.26%), and *Verrucomicrobia* (9.57%). The dominant phyla of gut microbiota in 0-day-old rabbits were *Proteobacteria* (67.27%), *Firmicutes* (8.89%), and *Actinobacteria* (6.04%). The dominant phyla of gut microbiota in 10-day-old rabbits included *Firmicutes* (38.57%), *Bacteroidetes* (30.48%), *Verrucomicrobia* (11.76%), and *ε Proteobacteria* (2.51%). The dominant phyla of gut microbiota in 60-day-old rabbits included *Firmicutes* (60.42%), *Bacteroidetes* (30.48%), and *Verrucomicrobia* (11.76%) (Figure 2C). The relative abundance of *Firmicutes* in rabbits gradually increased from birth to 60 days old, reaching the level of the female rabbits., while the relative abundance of *Proteobacteria* was gradually decreasing. At the genus level, the dominant genera of gut microbiota in female rabbits included *Akkermansia* (9.57%), *Bacteroides* (5.39%), and *Ralstonia* (1.62%). The dominant genera of gut microbiota in 0-day-old rabbits included *Ralstonia* (19.76%), *Enterococcus* (4.80%), and *Bibersteinia* (2.94%). The dominant genera of gut microbiota in 10-day-old rabbits included *Bacteroides* (24.30%), *Clostridium sensu stricto 1* (14.79%), and *Akkermansia* (11.76%). The dominant genera of gut microbiota in 60-day-old rabbits included *Akkermansia* (5.12%) and *Bacteroides* (1.61%) (Figure 2D). From birth to 60 days old, the abundance of gut microbiota gradually increased in rabbits, and then decreased with the introduction of solid feed.

### 3.3. Identification and Functional Prediction of Biomarkers for Female Rabbits and Their Different Aged Juvenile Rabbits

To further determine the impact of gut microbiota on function, LEfse analysis was used to identify significant biomarkers for different groups. Except for the female rabbit group, all groups contained identified biomarkers. As shown in Figure 3A, 32 biomarkers were identified in 0-day-old rabbits, mainly *Proteobacteria* and *Actinobacteria*. A total of 27 biomarkers were identified in 10-day-old rabbits, mainly belonging to the phylum *Campylobacter*, and 14 biomarkers were identified in 60-day-old rabbits, mainly belonging to the phylum *Firmicutes*. At the genus level, the genera *Achromobacter*, *Ralstonia*, and *Halomonas* were significantly different in 0-day-old rabbits, while the genera *Clostridium Sensu stricto 1* and *Epulopiscium* were significantly different in 10-day-old rabbits and the genera *Ruminococcus* and *Alistipes* were significantly different in 60-day-old rabbits (Figure 3B). In summary, we found that the microbial communities that play important roles in different growth stages of rabbits are different.

Tax4Fun functional prediction analysis was performed on the gut microbiota of female rabbits and their 0-, 10-, and 60-day-old rabbits to better understand the role of microbial communities in the host. There were significant differences in the metabolic pathways of KEGG Level 2 among juvenile rabbits of different ages (Figure 3C). The results showed that compared with the 10- and 60-day-old rabbits, the gut microbes of 0-day-old rabbits had high levels of biodegradation and metabolism, energy metabolism, vitamin metabolism, membrane transport, metabolism of infectious diseases, metabolism of polyketide compounds, lipid metabolism, and other amino acid metabolism functions. Compared to the 0- and 60-day-old rabbits, the gut microbiota of 10-day-old rabbits was enriched in folding, classification, degradation, cell aging, drug resistance, carbohydrate metabolism, polysaccharide biosynthesis, and metabolic functions. Compared to the 0- and 10-day-old rabbits, the gut microbiota of 60-day-old rabbits was enriched in transcriptional function. However, there was no significant functional difference in the gut microbiota of female rabbits compared to the 0- and 10-day-old rabbits.

### 3.4. The Influence of Gut Microbiota in Female Rabbits on the Establishment of Gut Microbiota in Offspring

To quantify the contribution of microorganisms from 36 female rabbits to the microbial community of 155 offspring rabbits, the FEAST algorithm was used. The results showed that the contribution of gut microbiota in female rabbits to the direct offspring microbiota was significantly higher (*p* < 0.01) than the mean contribution of other rabbits without blood ties (Figure 4A). Without blood ties refers to having no blood relation within two generations, with a minimum coefficient of relatedness of 0. The contribution of female rabbits to the gut microbiota of rabbits on the day of birth was the highest, reaching 23.52%. Compared to the day of birth, both the contribution to the 10-day-old (6.15%, *p* < 0.01) and the contribution to the 60-day-old (9.87%, *p* < 0.05) offspring significantly decreased. There was no significant difference in the contribution of female rabbits to the gut microbiota of 10-day-old rabbits and 60-day-old rabbits (Figure 4B).

### 3.5. Correlation Analysis between the Relative Abundance of Gut Microbiota and Body Weight in 60-Day-Old Rabbits

To estimate the correlation between gut microbiota strength and body weight in rabbits, firstly, the average weight of 35- and 60-day-old rabbits was calculated, which was 837.05 and 1701.69 g, respectively. Subsequently, a mixed linear model was used to compare the DIC values of various models. The results showed that the fitting effect continued to improve from model 1 to model 4, and the DIC value of model 4, which included pedigree information and relative microbial abundance as random variables, was the smallest. Compared with model 3, which only considered pedigree information, and model 2, which only considered relative microbial abundance, model 4 had a better degree of fit. The random error term gradually decreased from model 1 to model 4. The microbiability ranged from 0.15 ± 0.12 to 0.19 ± 0.14 (Table 1).

To screen for microorganisms significantly associated with body weight, six ASVs (ASV103, ASV308, ASV364, ASV598, ASV607, and ASV657) significantly associated with 60-day-old body weight (5% significance threshold) were identified through a single ASV regression analysis. Microbial group annotation was performed on these six ASVs, and it was found that all six ASVs belong to the family *Spirulinaceae*, among which three ASVs (ASV598, ASV607, ASV657) belong to the genus *Marvinbryantia* (Figure 5).

## 4. Discussion

This study conducted 16s rRNA sequencing analysis on the feces of female rabbits and their offspring at different ages to explore the changes in gut microbiota and observed differences in the types and relative abundance of gut microbiota among rabbits at different ages. The results of the alpha diversity analysis showed that the Shannon index of rabbits decreased first and then increased from birth to 60 days of age, and there were significant differences between groups (*p *< 0.05), indicating that the richness of gut microbiota in rabbits decreased first and then increased from birth to 60 days of age. The results of the beta diversity analysis showed that rabbits of different ages were dispersed at different positions in the PCoA space, but there was some spatial overlap between 60-day-old rabbits and female rabbits, indicating that the microbial community structure of 60-day-old rabbits was very close to that of female rabbits and tended to be stable. These were consistent with previous studies on rabbit cecal microbiota by Zhao et al. [19].

After classifying different bacterial groups, it was found that *Firmicutes*, *Bacteroidetes*, and *Verruciformes* were the dominant microorganisms in rabbits at different ages. Zhu et al. [20] found that the percentage of *Firmicutes* increased with the age of rabbits, which was consistent with our results. *Bacteroidetes* is considered one of the main symbiotic phyla in the rabbit intestine, which has been shown to stimulate the development of gut-related immune tissues [21,22]. Elisa et al. [23] found that *Verruciformes* was detected in relative abundance in all digestive tracts, especially in the large intestine. *Proteobacteria* was the most abundant microorganism in the gut microbiota of 0-day-old rabbits (67.27%), but its relative abundance gradually decreased with increasing rabbit age. Zhao et al. [18] found that *Proteobacteria* was the dominant phylum in the cecal microbiota of 2-day-old rabbits. Guzman et al. [24] detected the presence of *Proteobacteria* in the gut tract and feces of Holstein calves within less than 20 minutes of birth. Research has found that *Proteobacteria* was the most abundant not only in newborn animals but also in pregnant females and even prenatal embryos [25]. *Proteobacteria* was the dominant phylum in the uterus, vagina, gallbladder, and other areas of rabbits [26,27,28]. In summary, we speculate that the *Proteobacteria* phylum may originate from the female organism and gradually decrease with embryonic growth and development. More direct evidence of the specific *Proteobacteria* microbial origin from the female organism and the fact that the abundance gradually decreases with growth and development may require further systematic and in-depth research. The results of comprehensive Tax4Fun function prediction indicated that the early life (0- and 10-day-old) gut microbiota of rabbits was mainly enriched in metabolic levels, such as lipid metabolism, energy metabolism, and amino acid metabolism. These functional changes may be due to changes in microbial composition. Beaumont et al. [29] found that the relative concentrations of acetate, butyrate, propionate, glucose, and glutamate in the intestines of juvenile rabbits increased with age and the relative abundance of *Firmicutes* increased with age. Charlotte et al. [30] demonstrated that very early ingestion of solid food in infant rabbits, although in small quantities, induced changes in gut microbiota colonization and activity, with an acceleration of the ecological species succession and increased production of short-chain fatty acids.

In recent years, evidence from microbial community analysis has supported the vertical transmission of microbial communities from the female to the offspring [31,32]. Research has found that early colonization of gut microbiota is crucial for the health of piglets, and maternal birth canal, saliva, breast milk, litter environment, and feed were the main pathways for piglets to acquire microbial communities [33]. At present, there is relatively little research on the establishment of gut microbiota in juvenile rabbits by female rabbit gut microbiota, and research is limited to comparing the composition of gut microbiota between female and juvenile rabbits, lacking quantitative calculations for this impact. This study quantitatively calculated the specific contribution of female rabbits’ gut microbiota to the establishment of their juvenile rabbits’ gut microbiota using FEAST. In this study, although the contribution of female rabbits to newborn rabbits was the largest, it only reached 23.52%, and nearly 80% of the microbial sources were unknown, obtained either from the skin or from the mouth. The specific contributions of each source to gut microbiota may require further in-depth research. Although there were significant differences between female rabbits and newborn rabbits in phylum and genus level microbiota classification, the dominant bacteria at phylum and genus levels only included the top 10 in terms of abundance and did not include some low-abundance bacteria. These large numbers of low-abundance bacteria may have higher consistency in microbial community transmission in vertical transmission. Although 0- and 10-day-old rabbits were in the same environment, there may still be differences in the microbial community structure between the two groups due to differences in nutrient intake. 0-day-old rabbits were euthanized, and their nutritional sources were mainly due to umbilical cord transport during the embryonic stage. After birth, as the female rabbits breastfed, the main source of nutrition for the juvenile rabbits was breast milk. During the first few weeks after delivery, female rabbits excrete fecal pellets, which are ingested by their pups [34]. In addition, during the lactation process, the juvenile rabbits’ mouths came into contact with the female rabbits’ nipples, so a part of the gut microbiota may originate from the female rabbits’ skin. Similar to the previous description, this difference may require the measurement of the microbial community in multiple parts of the female rabbits’ nipple, mouth, vagina, skin, etc. for a more detailed explanation.

The slaughter weight of rabbits is an important economic trait, and changes in gut microbiota have long been considered one of the possible reasons for weight changes [35]. Previous studies have shown that daily weight gain under free feeding was a heritable trait, indicating that host genetics played an important role in regulating growth and feed utilization [36]. In addition, this study screened out six ASVs significantly correlated with body weight by correlating the relative abundance of microorganisms with the weight of 60-day-old rabbits. These ASVs were all microorganisms of the family *Lachnospiraceae*, which has a positive or negative statistical correlation with certain diseases. In these studies, changes in the abundance of *Lachnospiraceae* were more correlated than factors such as age, gender, and genetics [37]. In rabbits, we had found similar results. Fang et al. [38] found a significant correlation (*p *< 0.05) between the *Lachnospiraceae* family and the final body weight of fattened meat rabbits through two-part study models. At the genus level, we identified three ASVs in the *Marvinbryantia* genus that were significantly correlated with weight traits. Jing et al. [39] found that adding tobacco to the diet of rabbits increased their carcass weight, and the relative abundance of the *Marvinbryantia* genus significantly increased. Guo et al. [40] found in their study on the effects of high-fat feeding on rabbits that high-fat feeding increased the relative abundance of the *Marvinbryantia* genus, indicating that microbial dysregulation was related to lipid metabolism and inflammation. In summary, the microbial community of the *Marvinbryantia* genus under the classification of *Trichospiridae* may have a positive impact on the weight gain of rabbits, which may be due to the function of *Lachnospiraceae* in promoting the synthesis of short-chain fatty acids and promoting glucose and lipid metabolism. More systematic and in-depth research may require a combination of metabolomics and animal feeding experiments for validation.

## 5. Conclusions

This study determined the 16s rRNA of gut microbiota in rabbits of different age groups and preliminarily explored changes in the types of gut microbiota during the colonization process, as well as the impact of female rabbits’ gut microbiota on the colonization of gut microbiota in their juvenile rabbits. Six ASVs with a significant impact on weight were identified. They all belong to the family *Lachnospiraceae*. The research results provide a firm theoretical basis for maintaining gut health and constructing rabbit breeding methods.

## Figures and Tables

**Figure 1 animals-14-01741-f001:**
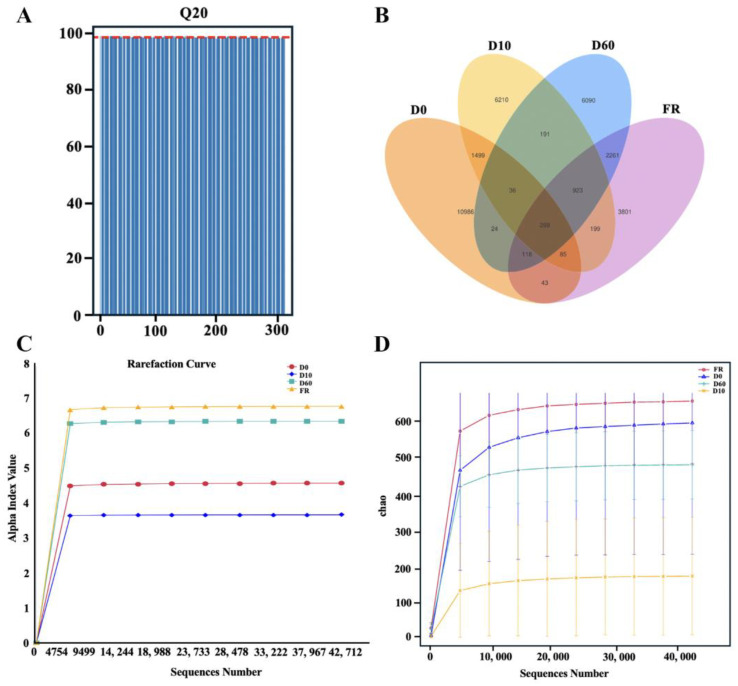
Data Evaluation: (**A**) Q20 level. (**B**) ASV Venn diagram for different groups. (**C**) Rarefaction curves for different groups. (**D**) Shannon curves for different groups. Abbreviations: FR, female rabbits; D0, 0-day-old rabbits; D10, 10-day-old rabbits; D60, 60-day-old rabbits.

**Figure 2 animals-14-01741-f002:**
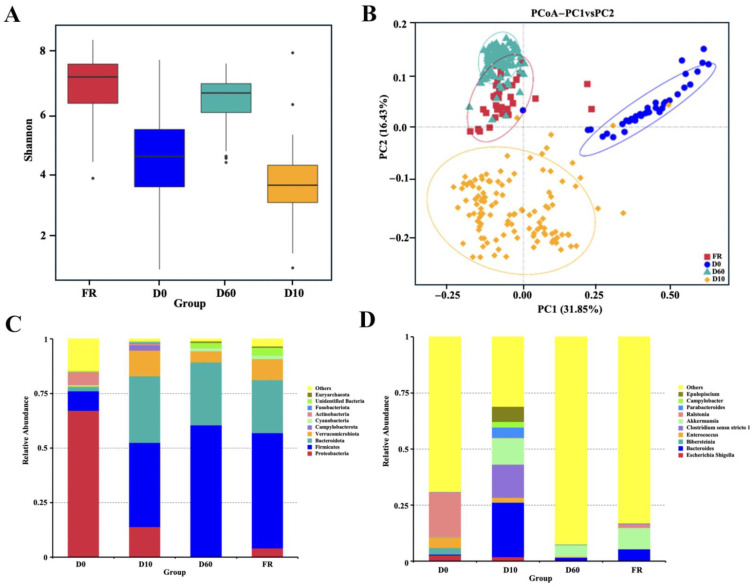
Composition and comparison of gut microbiota in female rabbits and their different aged juvenile rabbits: (**A**) Shannon diversity index of different groups. (**B**) Beta diversity of the genera analyzed by PCoA. (**C**) The relative abundance and distribution of the top 10 dominant phyla for different groups. (**D**) Relative abundance and distribution of the top 10 genera for different groups.

**Figure 3 animals-14-01741-f003:**
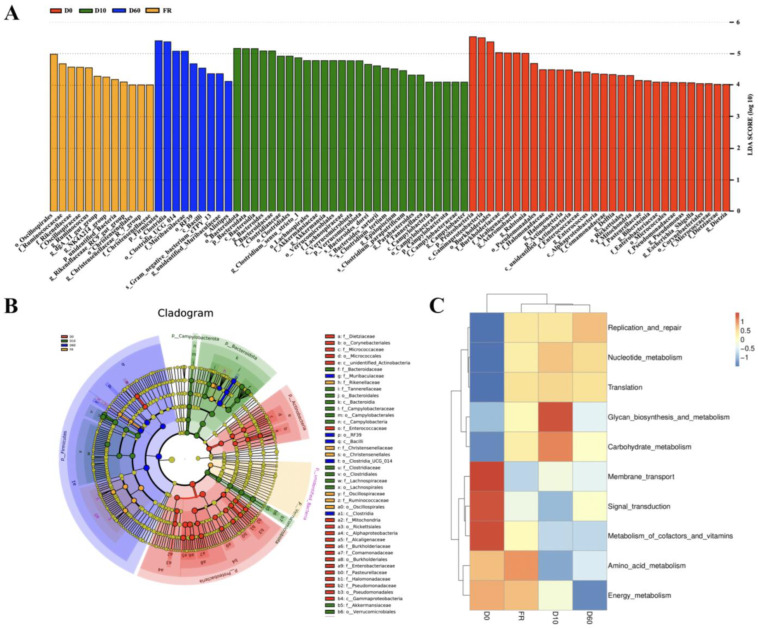
Identification and functional prediction of biomarkers for female rabbits and their different aged juvenile rabbits: (**A**) LEfSe analysis among different groups, with biomarkers determined by a linear discriminant analysis (LDA) score > 4. (**B**) Branching map of bacterial species with different abundance at different taxonomic levels. The root of the cladogram denotes the kingdom of bacteria, and the circle radiating from inside to outside represents the taxonomic levels from phylum to species. The size of each node represents relative abundance, and taxa with significant differences are painted with a designated color. (**C**) Heat map of significant differences in KEGG secondary metabolic pathways among groups.

**Figure 4 animals-14-01741-f004:**
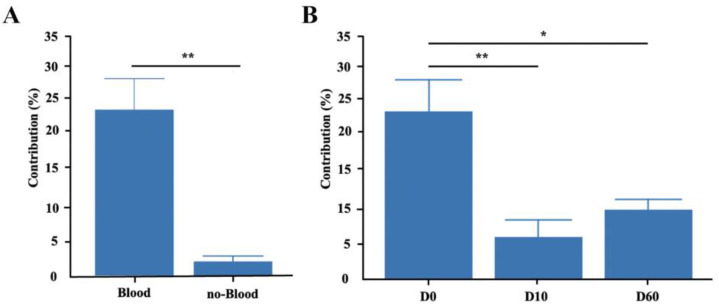
The influence of gut microbiota in female rabbits on the establishment of gut microbiota in offspring: (**A**) The contribution of maternal rabbit gut microbiota to different kinship groups. (**B**) Effect of female rabbits on the establishment of microbiota in offspring. The data are presented as means ± SEM. * *p* < 0.05; ** *p* < 0.01.

**Figure 5 animals-14-01741-f005:**
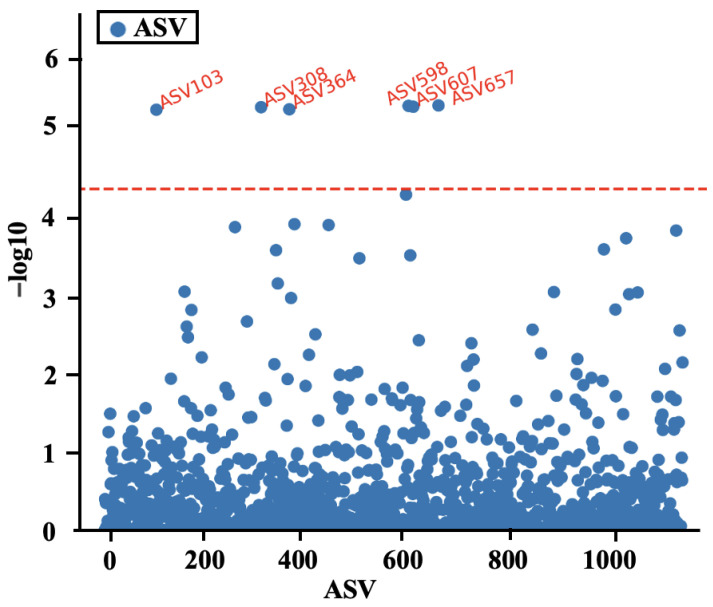
Results of microbiome-wide association analyses between amplicon sequence variant and 60-day weight. Dashed line represents significance and suggestive significance at 5% family-wise type I error rates.

**Table 1 animals-14-01741-t001:** Calculation of variance, heritability, microbiability posterior mean, and DIC.

	$\sigma_{g}^{2}$	$\sigma_{m}^{2}$	$\sigma_{e}^{2}$	$\sigma_{p}^{2}$	$H^{2}$	$M^{2}$	DIC
Model 1			840.99 ± 114.98	840.99 ± 114.98			1139.88
Model 2		176.07 ± 36.37	731.65 ± 148.54	907.72 ± 184.91	0.64 ± 0.17	0.19 ± 0.14	1082.73
Model 3	604.06 ± 217.91		338.12 ± 136.01	942.18 ± 152.93			1003.12
Model 4	657.45 ± 204.01	146.59 ± 11.26	197.08 ± 126.37	1001.12 ± 240.24	0.66 ± 0.15	0.15 ± 0.12	969.51

Note: $\sigma_{g}^{2}$ is random genetic variance. $\sigma_{m}^{2}$ is random microbiome variance. $\sigma_{e}^{2}$ is random residuals. $\sigma_{e}^{2}$ is the error term. $\sigma_{p}^{2}$ is 60 days body weight variance. $H^{2}$ is heritability. $M^{2}$ is microbiability. DIC is a deviance information criterion. Models 1, 2, 3, and 4 correspond to (1), (2), (3), and (4) in the Materials and Methods Section, respectively.

## Data Availability

All the figures and tables used to support the results of this study are included.

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
