# Peer review of "Study on Changes in Gut Microbiota and Microbiability in Rabbits at Different Developmental Stages"

_animals, 2024, doi:10.3390/ani14121741_

Round 1
Reviewer 1 Report
Comments and Suggestions for Authors
Overall, the manuscript presents a well-structured and detailed introduction. The objectives are well indicated. The experimental design and the description of material and methods could be improved. The results are precise and the discussion could be extended, as some results have not been discussed.
In particular:
Specifically:
section 2.2. Information on the diet of females and rabbits is missing. The collection days 0 and 60 are obvious for the purpose of the work, but a justification of why samples have been collected on day 10 is necessary. Why do the authors consider this age important?
Section 2.4 Define ASV and indicate the QIIME2 version used.
section 2.5. Pedigree matrix information is missing, number of animals?
line 255: ‘without blood ties’, please justify the minimum degree of relatedness considered.
section 3.5. The errors associated with model 2 and 4 estimates are too high to draw relevant conclusions. It should also be considered in the discussion (lines 345-348).
Line 352: in other studies the family Trichospiridae has not been relevant (https://microbiomejournal.biomedcentral.com/articles/10.1186/s40168-023-01580-4, https://doi.org/10.1186/s12711-024-00895-6).
Other:
. Check the use of ‘;’ throughout the text.
Author Response
Dear reviewer, thank you for taking the time out of your busy schedule to review this article. We have made the necessary changes based on your feedback. Please refer to the attachment for your review.

Reviewer 2 Report
Comments and Suggestions for Authors
The manuscript can be published after major revision considering the comments reported in the paper.
The main problem is that the manuscript has a very general introduction and discussion. The text should be more oriented to rabbits. Almost all the references are human, pig or poultry. There are very few references to rabbits. Several studies have been carried out on the microbiota of rabbits and its relationship with growth and intestinal health, which should be consulted and included in the references.
In the attached file you can find some more comments that need to be reviewed.

Comments on the Quality of English LanguageAuthor Response
Dear reviewer, thank you for taking the time out of your busy schedule to review this article. We have made the necessary changes based on your feedback. Please refer to the attachment for your review.

Reviewer 3 Report
Comments and Suggestions for Authors
See the attached file.

Comments on the Quality of English LanguageModerate editing of English language required.
Author Response

(The authors gave the same response as above.)

Round 2
Reviewer 2 Report
Comments and Suggestions for Authors
The manuscript has improved considerably, and can be published with minor revision:
In lines 71-72 of the Introduction, it is indicated that "there have been no reports on rabbits to support this conclusion" and it is not entirely true because there are some previous works that have studied this. I show you some references that may be useful to include at this point in the introduction and in the discussion.
· DOI: 10.1128/msystems.00243-22
Early Introduction of Plant Polysaccharides Drives the Establishment of Rabbit Gut Bacterial Ecosystems and the Acquisition of Microbial Functions
· DOI: 10.1093/jn/nxab411
Developmental Stage, Solid Food Introduction and Suckling Cessation Differentially Influence the Co-maturation of the Gut Microbiota and Intestinal Epithelium in Rabbits
· DOI: 10.1017/S1751731120001305
Evolution of gut microbial community through reproductive life in female rabbits and investigation of the link with offspring survival
· DOI: 10.2527/jas2013-6394
Coprophagous behavior of rabbit pups affects implantation of cecal microbiota and health status
· DOI: 10.1016/j.anifeedsci.2011.12.016
Nutritional digestive disturbances in weaner rabbits
· DOI: 10.4995/wrs.2017.5230
Pyrosequencing study of caecal bacterial community of rabbit does and kits from a farm affected by epizootic rabbit enteropathy
Author Response
Dear reviewer, thank you for taking the time out of your busy schedule to review this article. Thank you for all your help with this article. I have made slight modifications to the introduction and discussion sections. Please refer to the attachment for details. Thank you!

Reviewer 3 Report
Comments and Suggestions for Authors
I can accept the revised manuscript in present form.
Author Response
Dear reviewer, thank you for taking the time out of your busy schedule to review this article. Thank you for all your help with this article.